# Infrared characterization of formation and resonance stabilization of the Criegee intermediate methyl vinyl ketone oxide

Chen-An Chung[1] & Yuan-Pern Lee [ID] [1,2,3 ✉]

Methyl vinyl ketone oxide (MVKO) is an important Criegee intermediate in the ozonolysis of isoprene. MVKO is resonance stabilized by its allyl moiety, but no spectral characterization of this stabilization was reported to date. In this study, we photolyzed a mixture of 1,3-diiodo-but-2-ene and $O_2$ to produce MVKO and characterized the *syn-trans*-MVKO, and tentatively *syn-cis*-MVKO, with transient infrared spectra recorded using a step-scan Fourier-transform spectrometer. The O–O stretching band at $948\ cm^{-1}$ of *syn–trans*-MVKO is much greater than the corresponding bands of *syn*-$CH_3CHOO$ and $(CH_3)_2COO$ Criegee intermediates at 871 and $887\ cm^{-1}$, respectively, confirming a stronger O–O bond due to resonance stabilization. We observed also iodoalkenyl radical $C_2H_3C(CH_3)I$ upon photolysis of the precursor to confirm the fission of the terminal allylic C–I bond rather than the central vinylic C–I bond of the precursor upon photolysis. At high pressure, the adduct $C_2H_3C(CH_3)IOO$ was also observed. The reaction mechanism is characterized.

[1] Department of Applied Chemistry and Institute of Molecular Science, National Chiao Tung University, 300093 Hsinchu, Taiwan. [2] Center for Emergent Functional Matter Science, National Chiao Tung University, 300093 Hsinchu, Taiwan. [3] Institute of Atomic and Molecular Sciences, Academia Sinica, 106319 Taipei, Taiwan. ✉email: yplee@nctu.edu.tw

Methane ($CH_4$) and isoprene [$CH_2$=CH–C($CH_3$)=$CH_2$] are the two most abundant volatile organic compounds (VOC) emitted into Earth's atmosphere. The dominant atmospheric removal paths of isoprene, which has a total global emission of about 500–750 Tg year$^{-1}$[1,2], are its reactions with OH, $NO_3$, and ozone ($O_3$)[3–5]. Ozone is responsible for the removal of ~10% of isoprene[6] and the associated formation of OH in the atmosphere[7]. Similar to the ozonolysis of alkene, the ozonolysis of isoprene produces Criegee intermediates; in this case three Criegee intermediates formaldehyde oxide ($CH_2OO$), methyl vinyl ketone oxide [MVKO, $C_2H_3C(CH_3)OO$], and methacrolein oxide [MACRO, $CH_2$=C($CH_3$)CHOO] are produced with branching ratios ~58%, 23%, and 19%, respectively[6,8,9]. The decomposition of some internally excited Criegee intermediates might yield OH. Recent measurements indicate that the yield of OH from the reaction isoprene + $O_3$ is about 25 %[6,10,11]. MVKO and MACRO are important Criegee intermediates, not only because of their critical roles in atmospheric chemistry, but also because of their unique structures that are resonance stabilized by the vinyl moiety, which affect their reactivity.

Welz et al.[12] reported a novel reaction scheme to generate the simplest Criegee intermediate $CH_2OO$ in laboratories from the reaction of $CH_2I$ with $O_2$ on photolysis of $CH_2I_2$ in $O_2$ with ultraviolet (UV) light; this scheme has promoted active research on Criegee intermediates[13–18]. However, following this concept, to produce MVKO from photolysis of $C_2H_3C(CH_3)I_2$ in $O_2$, is difficult because this precursor is extremely unstable. Barber et al. recently reported a novel method to produce MVKO on photolysis of 1,3-diiodo-but-2-ene [($CH_2I$)HC=C($CH_3$)I, (1) in Fig. 1]

and $O_2$ in a gaseous mixture with UV light[19]. Four conformers of MVKO, *syn-trans*, *syn-cis*, *anti-trans*, and *anti-cis* (shown in Supplementary Fig. 1) are predicted to exist; the *syn-* and *anti-* indicate the orientation of the methyl group relative to the terminal O atom, and *cis-* and *trans-* indicate the relative direction of the terminal C=C bond and the C=O bond. Barber et al. characterized *syn-* and *anti-*MVKO utilizing near-infrared (NIR) action spectra, in which NIR activation of MVKO in region 5750–6260 $cm^{-1}$ resulted in dissociation of MVKO to produce OH, which was detected with laser-induced fluorescence[19]. Using the same production scheme, Vansco et al.[20] and Caravan et al.[21] reported the UV spectrum of MVKO. Recently, the microwave spectrum of *syn-trans*-MVKO was reported[22].

Barber et al. assumed that photolysis of (1) at 248 nm resulted in a preferential dissociation of the terminal allylic C–I bond, rather than the vinylic C–I bond, to form the iodoalkenyl radical 3-iodo-but-2-en-1-yl [$C_2H_3C(CH_3)I$, (2) in Fig. 1]; a subsequent addition reaction of $O_2$ with (2), followed by breaking the remaining C–I bond, produced MVKO. However, gaseous (2) has never been spectrally characterized. The mid-infrared spectrum of MVKO and other related intermediates such as (2) are expected to provide definitive characterization of the conformation of MVKO, evidence for the resonance stabilization, and valuable information on the detailed mechanism for the source reaction.

With the unique step-scan Fourier-transform infrared (FTIR) absorption technique[23], we have successfully detected infrared spectra of several smaller Criegee intermediates $CH_2OO$[24,25], $CH_3CHOO$[26], $(CH_3)_2COO$[27], and the associated adduct $CH_2IOO$ from the source reaction[28]; we also explored the mechanism and

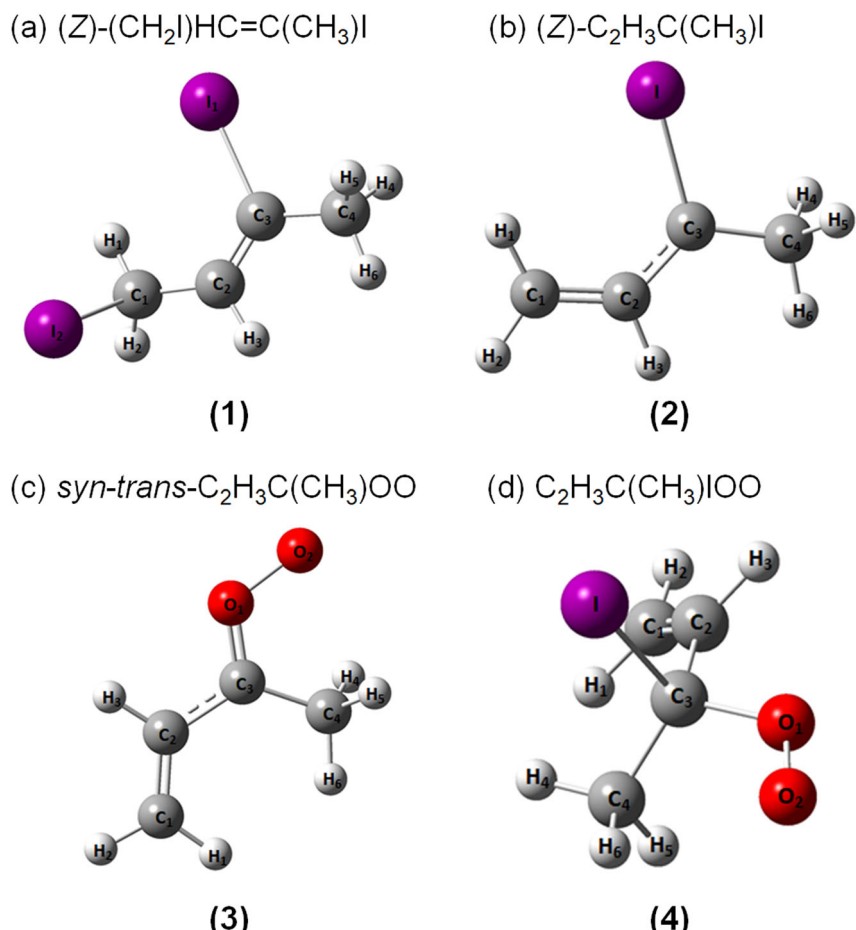

(a) (Z)-($CH_2I$)HC=C($CH_3$)I

(b) (Z)-$C_2H_3C(CH_3)I$

(1)

(2)

(c) *syn-trans*-$C_2H_3C(CH_3)OO$

(d) $C_2H_3C(CH_3)IOO$

(3)

(4)

**Fig. 1 Geometries of species observed in this work. a** Precursor (Z)-($CH_2I$)HC=C($CH_3$)I **(1)**. **b** Iodoalkenyl radical (Z)-$C_2H_3C(CH_3)I$ **(2)**. **c** Criegee intermediate *syn-trans*-$C_2H_3C(CH_3)OO$ **(3)**. **d** Iodoperoxy radical $C_2H_3C(CH_3)IOO$ **(4)** predicted with the B3LYP/aug-cc-pVTZ-pp method.

intermediates in reactions of $CH_2OO$ with $CH_2OO$[29], $SO_2$[30], and HCOOH[31]. Advantages of this technique include its wide spectral and temporal coverage so that most reaction intermediates were probed simultaneously and the detailed reaction mechanism might be unraveled. In this work we extended our focus to a larger and atmospherically important Criegee intermediate. UV photodissociation of the precursor (1) yields the formation of (2), confirming that only the terminal allylic C–I bond was dissociated. When oxygen at varied pressure was added, the IR spectra of two conformers syn-trans-MVKO [$C_2H_3C(CH_3)OO$, (3) in Fig. 1] and syn-cis-MVKO, and the adduct 3-iodo-but-1-en-3-yl-peroxy [$C_2H_3C(CH_3)IOO$, (4) in Fig. 1] radical were distinctly identified; the mechanism of the source reaction is characterized.

## Results and discussion

**Observation of the iodoalkyl radical (Z)-$C_2H_3C(CH_3)I$ (2).** We employed the B3LYP/aug-cc-pVTZ-pp method to characterize all species of interest. The optimized geometries and vibrational wavenumbers of these species, including a possible cyclic peroxide product dioxole[21], are presented in Supplementary Note 1, including Supplementary Figs. 1–4 and Supplementary Tables 1–10. The geometries of major species observed in this work—the precursor (Z)-$(CH_2I)HC=C(CH_3)I$ (1), the iodoalkenyl radical (Z)-$C_2H_3C(CH_3)I$ (2), Criegee intermediate syn-trans-$C_2H_3C(CH_3)OO$ (3), and iodoperoxy radical $C_2H_3C(CH_3)IOO$ (4) are depicted in Fig. 1.

The precursor 1,3-diiodo-but-2-ene $(CH_2I)HC=C(CH_3)I$ is predicted to exist in (Z)- and (E)-conformations, with the former (1) 7.3 kJ mol$^{-1}$ more stable than the latter. We employed pure (Z)-conformer in this experiment; its IR spectral characterization is described in Supplementary Note 2 and shown in Supplementary Fig. 5. When the diiodoalkene precursor (1) in $N_2$ was irradiated with light at 248 nm, we observed six features near 1406, 1261, 1109, 1019, 925, and 873 cm$^{-1}$, as discussed in Supplementary Note 3 and shown in Supplementary Fig. 6. We termed these six features that are associated with the primary photolysis product as group A and marked them $A_1$–$A_6$ in Fig. 2a. The stick spectra of two possible photolysis products, (Z)-$C_2H_3C(CH_3)I$ (2) and (Z)-$(CH_2I)CHC(CH_3)$, according to the scaled harmonic vibrational wavenumbers predicted with the B3LYP method are shown in Fig. 2b and c, respectively. The observed new features agree satisfactorily with lines predicted near 1418, 1261, 1108, 1018, 930, and 887 cm$^{-1}$ for (2), as compared in Fig. 2 and Supplementary Table 11; they agree poorly with the spectrum predicted for (Z)-$(CH_2I)CHC(CH_3)$, shown in Fig. 2c, and (E)-$C_2H_3C(CH_3)I$, shown in Supplementary Fig. 7. Observation of (Z)-$C_2H_3C(CH_3)I$ confirms that the terminal C–I bond was broken upon irradiation at 248 nm and that an interconversion between the (Z)- and (E)-conformers did not occur. We could not, however, definitely exclude the formation of (Z)-$(CH_2I)CHC(CH_3)$ in a small proportion because, in region 1450–850 cm$^{-1}$, the only intense line predicted for this species is near 1146 cm$^{-1}$ (Fig. 2c), which might overlap with the intense band of the precursor near 1152 cm$^{-1}$; hence the upper limit for the percentage of production of (Z)-$(CH_2I)CHC(CH_3)$ could not be estimated. However, according to our previous results on photolysis of (Z)-$(CH_2I)HC=C(CH_3)I$ in solid $p$-$H_2$ at 290 nm, which cover the entire IR region to include several other intense features of the products, the formation of (Z)-$(CH_2I)CHC(CH_3)$ on breaking the central C–I bond was unobserved[32].

## Observation of the Criegee intermediate $C_2H_3C(CH_3)OO$ (3).

When precursor (1) (0.04 Torr) and $O_2$ (35 Torr) was irradiated with light at 248 nm, a set of new features (group B) appeared and

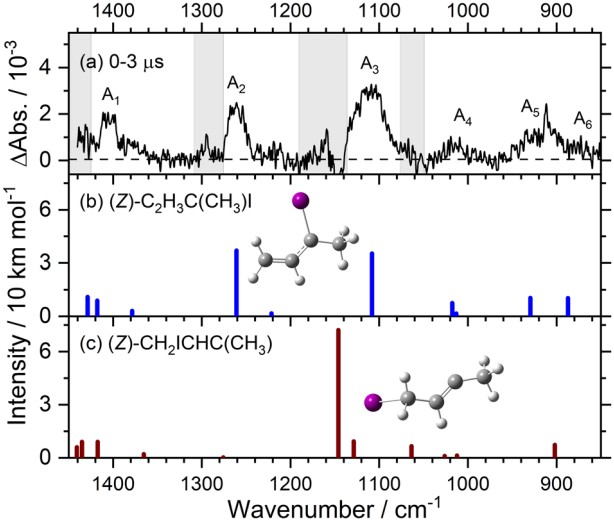

**Fig. 2 Comparison of bands in group A with simulated spectra of two isomers of iodoalkenyl radical. a** Spectrum (resolution 1.0 cm$^{-1}$) of bands in group A recorded 0–3 μs after photolysis; taken from Supplementary Fig. 6(e); bands are labeled $A_1$–$A_6$. Gray areas represent regions of possible interference from absorption of the parent molecule. IR stick spectrum predicted for (Z)-$CH_2CHC(CH_3)I$ (2) and (Z)-$(CH_2I)CHC(CH_3)$ according to scaled harmonic vibrational wavenumbers and IR intensities predicted with the B3LYP/aug-cc-pVTZ-pp method are shown in **b** and **c**, respectively.

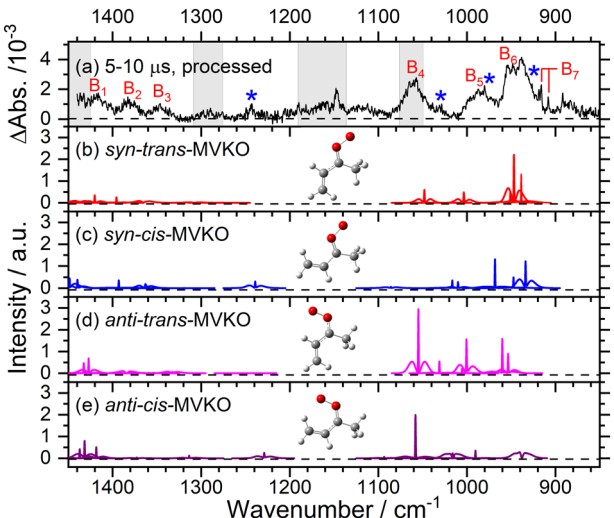

**Fig. 3 Comparison of bands in group B with simulated spectra of various conformers of Criegee intermediate MVKO. a** Spectrum (resolution 0.5 cm$^{-1}$) of bands in group B recorded 5–10 μs after photolysis; taken from Supplementary Fig. 8(g); bands are labeled $B_1$–$B_7$. Gray areas represent regions of possible interference from absorption of the parent molecule. IR spectra simulated with PGOPHER ($J_{max} = 200$, $T = 298$ K, FWHM = 0.64 cm$^{-1}$) are shown for **b** syn-trans-MVKO (3), **c** syn-cis-MVKO, **d** anti-trans-MVKO, and **e** anti-cis-MVKO. Possible bands of syn-cis-MVKO are marked with blue asterisks.

reached their maxima near 5–10 μs, while bands of the end product methyl vinyl ketone (MVK, $C_2H_3C(O)CH_3$) appeared at a later period and became more prominent in the spectrum recorded 30–35 μs after irradiation, as detailed in Supplementary Note 4 and Supplementary Fig. 8. The spectrum of bands in group B is presented in Fig. 3a to compare with the predicted IR stick spectra of four possible conformers of the Criegee

intermediates *syn-trans-*, *syn-cis-*, *anti-trans-*, and *anti-cis-*MVKO, Fig. 3b–e; we employed anharmonic vibrational wavenumbers and rotational parameters predicted with the B3LYP method and simulated the rotational contours with the PGO-PHER program[33], as described in Supplementary Note 5.

The observed spectrum matches best with the spectrum simulated for *syn-trans-*MVKO (**3**) in terms of vibrational wavenumbers, relative intensities, and rotational contours. The observed vibrational wavenumbers near 1416, 1383, 1346, 1060, 987, 948, and 908/916 cm$^{-1}$ agree satisfactorily with the anharmonic vibrational wavenumbers predicted near 1420, 1396, 1365, 1048, 1004, 947, and 939 cm$^{-1}$ for (**3**), as compared in Supplementary Table 12. The average absolute deviation between experiments and anharmonic vibrational calculations of *syn-trans-*MVKO (**3**), 12.7 ± 7.9 cm$^{-1}$, is much smaller thaan those for *syn-trans-*, *syn-cis-*, *anti-trans-*, and *anti-cis-*MVKO, 23.6 ± 13.7, 23.4 ± 18.3, and 42.6 ± 46.8 cm$^{-1}$, respectively; the listed errors represent one standard deviation in fitting. The corresponding average deviations for scaled harmonic vibrational calculations are 16.6 ± 10.8, 24.9 ± 14.2, 25.9 ± 18.5, and 33.0 ± 28.7 cm$^{-1}$, respectively.

The most intense band (O–O stretching mode, B$_6$) near 948 cm$^{-1}$ shows a clear rotational contour with *P-*, *Q-*, *R-*branches; the width of the contour is greater than predicted, presumably because the product is internally hot, as was observed previously for Criegee intermediates CH$_3$CHOO[26] and (CH$_3$)$_2$COO[27]. Furthermore, the intense, sharp *c*-type band of the CH$_2$-wagging ($\nu_{25}$) mode in this region was predicted to be near 939 cm$^{-1}$, which corresponds well with the most intense one observed at 916 cm$^{-1}$ (B$_7$) among several sharp lines. Because of the prominent *Q*-branch of this transition, the additional weaker line observed at 908 cm$^{-1}$ might be assigned to a hot band of this transition.

A small contribution of *syn-cis-*MVKO to the observed spectrum might be present, but we are unable to confirm this definitively because of its small intensity. Possible bands of *syn-cis-*MVKO are marked with * in Fig. 3a and compared with calculations in Supplementary Table 13. The small sharp feature at 980 cm$^{-1}$ (overlapped with band B$_5$) might correspond to the most intense *c*-type line of *syn-cis-*MVKO predicted near 968 cm$^{-1}$. The intense OO-stretching mode predicted near 934 cm$^{-1}$ might be overlapped with the *P*-branch of the broad feature B$_6$ of *syn-trans-*MVKO (**3**) near 948 cm$^{-1}$. Small features near 1243 and 1031 cm$^{-1}$ also match with predicted wavenumbers of 1239 and 1017 cm$^{-1}$, respectively. The *cis-* and *trans-*conformers were reported to interconvert rapidly by rotation about the C–C bond at 298 K because of the small barriers[21]. The energy gap between *syn-trans-* and *syn-cis-*MVKO, ~7 kJ mol$^{-1}$, implies that the latter has a population only ~6% of the former, according to a Boltzmann distribution at 298 K; our observation of predominant *syn-trans-*MVKO with a likely small proportion of *syn-cis-*MVKO is consistent with this distribution.

We compare the key geometries and vibrational wavenumbers of (**3**) with *syn-*CH$_3$CHOO, *anti-*CH$_3$CHOO, and (CH$_3$)$_2$COO to understand the resonance stabilization of MVKO. The lengths of the O–O and C–O bonds of these four species are compared in Table 1. The O–O length of 1.353 Å for (**3**) is significantly smaller than those of alkyl Criegee intermediates (~1.380 Å), whereas the C–O length of 1.297 Å for (**3**) is slightly longer than other species (1.270–1.284 Å). Spectroscopically, the OO-stretching vibrational wavenumbers of 948 cm$^{-1}$ for (**3**) is much greater than the corresponding values 871.2–887.4 cm$^{-1}$ for other species; the 948 cm$^{-1}$ is the largest OO-stretching wavenumbers for all Criegee intermediates observed so far. Going from CH$_2$OO to *anti-*CH$_3$CHOO and (CH$_3$)$_2$COO, upon the replacement of the H atom(s) with CH$_3$, the OO-stretching wavenumber decreased slightly because of the electron-donating character of the CH$_3$ group enhances the C–O bond and weakens the O–O bond

**Table 1 Comparison of bond lengths and OO-stretching vibrational wavenumbers of Criegee intermediates.**

| | MVKO (3)[a] | *syn-*CH$_3$CHOO[b] | *anti-*CH$_3$CHOO[b] | (CH$_3$)$_2$COO[c] |
|---|---|---|---|---|
| r(O–O)/Å | 1.353 | 1.380 | 1.381 | 1.380 |
| r(C–O)/Å | 1.297 | 1.284 | 1.279 | 1.270 |
| $\nu$(OO)/cm$^{-1}$ | 948 | 871 | 884 | 887 |

[a]Bond distances predicted with the CCSD(T)/cc-pVTZ method; ref. [22].
[b]Bond distances predicted with the NEVPT2(8,8)/aug-cc-pVDZ method; ref. [26].
[c]Bond distances predicted with the B3LYP/aug-cc-pVTZ method; ref. [27].

slightly. The OO-stretching wavenumber of *syn-*CH$_3$CHOO decreases further because of the interaction of the terminal O atom with the two H atoms in CH$_3$ weakens the O–O bond. The much larger OO-stretching wavenumber of MVKO is due to the resonance stabilization. The resonance stabilization can be comprehended from two major resonance structures with conjugated double bonds, shown in Supplementary Fig. 9(a); the frontier orbitals of delocalized electrons with 0–2 nodes are shown in Supplementary Fig. 9(b). All these evidences support that the COO moiety of MVKO is resonance stabilized by the adjacent vinyl group so that the delocalization strengthens the O–O bond significantly as compared with alkyl Criegee intermediates.

Reported reaction kinetics of MVKO is so far limited, but a much slower removal of MVKO by water has been reported; this small reactivity was attributed to a higher energy of the transition state arising from the disruption of the extended conjugation of MVKO in reaction leading to the hydroperoxide adduct[21].

**Observation of the iodoperoxy radical C$_2$H$_3$C(CH$_3$)IOO (4).** When precursor (**1**) (0.042 Torr) and O$_2$ (347 Torr) was irradiated at 248 nm, a set of new features in group C, shown in Fig. 4a, was identified, as discussed in Supplementary Note 6 and shown in Supplementary Figs. 10–13. For the simplest Criegee intermediate CH$_2$OO, both CH$_2$OO and the adduct CH$_2$IOO were produced from photolysis of CH$_2$I$_2$ and O$_2$; the yield of CH$_2$OO decreased as the pressure increased because of the stabilization of the adduct at high pressure[28]. Possible structures of the adducts of the source reaction (**2**) + O$_2$ are hence C$_2$H$_3$C(CH$_3$)IOO (9 conformers, Supplementary Fig. 3) or, less likely, C(CH$_3$)ICHCH$_2$OO (6 conformers, Supplementary Fig. 4), with O$_2$ added to the carbon atom on either side of the allyl moiety.

Bands of group C are compared with the predicted stick spectra of the two least-energy conformers of C$_2$H$_3$C(CH$_3$)IOO and the least-energy conformer of C(CH$_3$)ICHCH$_2$OO, shown in Fig. 4b–d. The observed C$_1$–C$_6$ bands near 1375, 1213, 1108, 1063, 986, and 885 cm$^{-1}$ agree satisfactorily with the scaled harmonic vibrational wavenumbers predicted for both conformers of C$_2$H$_3$C(CH$_3$)IOO, Supplementary Table 14; they are expected to have a major contribution to the observed spectrum, even though we cannot assign definitively the conformation of the carriers for the observed spectrum. The agreement between observed bands and predicted spectrum of C(CH$_3$)ICHCH$_2$OO, Fig. 4d, is much less satisfactory. The facts that C$_2$H$_3$C(CH$_3$)I (**2**) was produced upon photolysis of the precursor and that C$_2$H$_3$C(CH$_3$)OO (**3**) was observed when O$_2$ was present at low pressure also support the assignment of observed bands in group C to C$_2$H$_3$C(CH$_3$)IOO (**4**).

**Temporal profiles of observed species.** An advantage of IR absorption is that each species has its characteristic absorption bands that can be identified and monitored during a reaction. Even though the three radical species observed in this work have

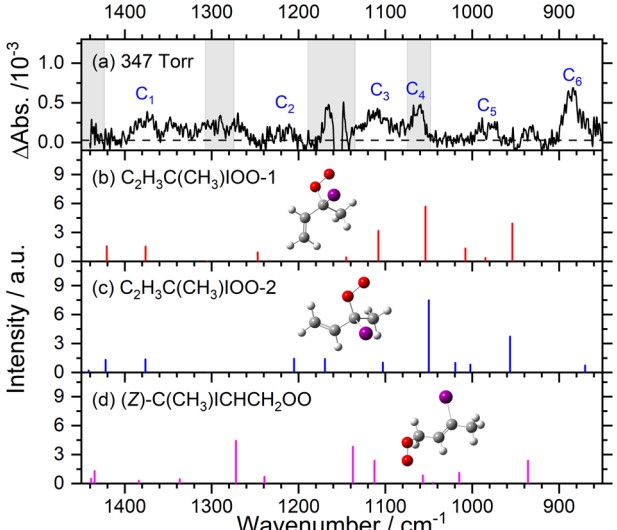

**Fig. 4 Comparison of bands in group C with predicted stick spectra of various isomers of iodoperoxy radicals. a** Experimental spectrum at 347 Torr, taken from Supplementary Fig. 10(g); bands of group C are labeled. Grey areas represent regions of possible interference from absorption of the precursor. IR stick spectra according to scaled harmonic vibrational wavenumbers and IR intensities predicted with the B3LYP/aug-cc-pVTZ-pp method are shown for two least-energy conformers of $C_2H_3C(CH_3)OO$ (**b, c**) and the least-energy conformer $(Z)$-$C(CH_3)ICHCH_2OO$ (**d**).

a similar base structure $C_2H_3C(CH_3)$, we were able to locate some absorption regions that are unique to each species and less susceptible to interference from other species.

The temporal profiles of **(2)**, **(3)**, **(4)**, and MVK (Supplementary Fig. 14) are discussed in Supplementary Note 7. The similar rates of rise for MVKO **(3)** and the adduct **(4)** and the rate of decay of **(2)** are consistent with the expectation from a parallel reaction, supporting that both species were produced from the same reaction, **(2)** with $O_2$. In contrast, the temporal profile of MVK has a slower rise, indicating the nature of a secondary reaction. Estimates of relative yields of **(3)** and **(4)** are presented in Supplementary Note 8. From Supplementary Table 15, the relative yield of **(3)** decreased to ~63%, whereas that of **(4)** increased significantly by a factor 1.8–2.9, as the pressure of $O_2$ increases from 35 Torr to 347 Torr; this observation confirms that more iodoperoxy adduct is stabilized at high pressure.

Based on temporal profiles of varied species during the reaction, one would be able to obtain information such as rate coefficients of formation, the relative yields of MVKO **(3)** and $C_2H_3C(CH_3)IOO$ **(4)** as a function of pressure, and kinetics on reactions of MVKO **(3)** with atmospheric species. More careful experiments under varied conditions are needed

## Conclusion

Upon irradiation of gaseous precursor $(Z)$-1,3-diiodo-but-2-ene **(1)** with light at 248 nm, iodoalkenyl radical $(Z)$-$C_2H_3C(CH_3)I$ **(2)** was produced, confirming that only the terminal allylic C–I bond, not the central vinylic C–I bond, was dissociated upon photolysis. When $O_2$ at 35 Torr was added to the system, Criegee intermediate MVKO, $C_2H_3C(CH_3)OO$, was identified; the spectrum agrees best with that predicted for the *syn-trans*-conformer **(3)**, but the *syn-cis*-MVKO might be present in a small proportion. With $O_2$ at 80–347 Torr, iodoperoxy radical adduct $C_2H_3C(CH_3)IOO$ **(4)** was observed, confirming the stabilization of the adduct at high pressure. The OO-stretching band at 948 cm$^{-1}$ is much greater than other reported Criegee intermediates, confirming that MVKO is resonance stabilized by its allyl moiety. To the best of our

knowledge, the IR spectra of these three intermediates are new; they provide valuable information to probe these species to understand the kinetics and mechanism of formation of Criegee intermediate MVKO from the source reaction and reactions of MVKO with atmospheric species in laboratories.

## Methods

**Experimental methods**. Details of the step-scan Fourier-transform infrared (FTIR) absorption spectrometer employed in this work are described elsewhere[23,24]. A White cell of volume ~1370 cm$^3$ and effective path length 3.6 m (base length 15 cm) served as a reactor and an absorption cell; it was coupled to a step-scan FTIR spectrometer via its external port. A KrF excimer laser (248 nm, 8 Hz, ~220 mJ pulse$^{-1}$, beam size 2.8 × 1.1 cm$^2$) was employed for the photo-dissociation of $(Z)$-$(CH_2I)HC{=}C(CH_3)I$ **(1)**. The photolysis laser beam was multiply reflected between a pair of external laser mirrors and propagated sideways, nearly perpendicular to the IR beams in the White cell.

The IR probe beam from the FTIR instrument was detected with a HgCdTe detector at 77 K with *dc*- and *ac*-coupled outputs. These signals were sent to an external 14-bit digitizer at a digitization intervals of 4 ns; typically, 10,000 data points were acquired to cover a period of 40 µs. Sometimes an internal 24-bit digitizer with temporal resolution 12.5 µs was used to cover a longer period and provide an improved ratio of signal to noise. To decrease the period of data acquisition, we employed appropriate optical filters and performed undersampling. For a spectral range 850–1450 cm$^{-1}$ at instrumental resolution 1 cm$^{-1}$, 1523 scan steps (each averaged with 15 laser shots) were completed in ~50 min; for resolution 0.5 cm$^{-1}$, 2843 scan steps were completed in ~90 min. The spectral width (full width at half maximum, FWHM) after apodization with the Blackman–Harris 3-term function is 1.28 times the listed instrumental resolution. To yield a spectrum with a satisfactory ratio of signal to noise, 4–14 spectra under similar conditions were accumulated and averaged.

The liquid sample of $(Z)$-$(CH_2I)HC{=}C(CH_3)I$ **(1)** was placed in a dark flask at 298 K; a stream of gaseous $O_2$ was passed over the sample to carry the vapor into the reactor. With a laser fluence of ~$8.9 \times 10^{16}$ photons cm$^{-2}$, the average photolysis efficiency of **(1)** was estimated to be ~13% according to its decrease in infrared absorbance. Based on this value and a ratio of 2.65 for the IR probed volume to the UV-photolysis volume determined previously[31], we estimated the absorption cross section of **(1)** at 248 nm to be $(4.0 \pm 2.0) \times 10^{-18}$ cm$^2$ molecule$^{-1}$. The decrease of the precursor upon irradiation was estimated to be $(1.1–3.4) \times 10^{14}$ molecule cm$^{-3}$. The partial pressures of **(1)** were derived on comparing the observed integrated absorbance of IR bands in regions 1130–1190 cm$^{-1}$ and 1025–1085 cm$^{-1}$ with the calibration curve obtained at varied pressures. The flow rate of $O_2$ was $F_{O_2} \cong 18.3$–41.7 STP cm$^3$ s$^{-1}$ (STP denotes standard temperature 273 K and pressure 1 atm). Partial pressures were $P_{(CH_2I)HC{=}C(CH_3)I} \cong 0.025$–0.050 Torr and $P_{O_2} = 35$–347 Torr. $(Z)$-$(CH_2I)HC{=}C(CH_3)I$ (>95%, Accela ChemBio), and $O_2$ (99.99%, Chiah-Lung) were used as received.

**Computational methods**. Quantum-chemical calculations were performed with the Gaussian 16 program[34]. We employed the B3LYP density-functional theory (DFT), which uses Becke's three-parameter hybrid exchange functional with a correlation functional of Lee et al.[35–37] to investigate the equilibrium geometry, rotational parameters, harmonic vibrational wavenumbers, and IR intensities of precursor $(CH_2I)HC{=}C(CH_3)I$, isomers of iodoalkenyl $C_2H_3C(CH_3)I$ and $(CH_2I)CHC(CH_3)$ radicals, four isomers of MVKO, and the iodoperoxy adducts $C_2H_3C(CH_3)IOO$ and $C(CH_3)ICHCH_2OO$. The anharmonic vibrations were calculated only for isomers of MVKO with a second-order perturbation approach using an effective finite-difference evaluation of the third and semi-diagonal fourth derivatives; MVKO contains no I atom so that anharmonic vibrational calculations are feasible for molecules of this size. The standard Dunning's correlation-consistent basis set augmented with diffuse functions, aug-cc-pVTZ, was used[38,39]. For the iodine atom, the additional pseudopotential, indicated as pp, was implemented[40].

## Data availability

The data supporting the findings of this study are available within the paper and its Supplementary Information. All other relevant data are available from the authors upon reasonable request.

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

## Acknowledgements

This work was supported by Ministry of Science and Technology, Taiwan (grants MOST108-2639-M009-001-ASP and MOST108-3017-F009-004) and the Center for Emergent Functional Matter Science of National Chiao Tung University from The Featured Areas Research Center Program within the framework of the Higher Education Sprout Project by the Ministry of Education (MOE) in Taiwan. The National Center for High-Performance Computation provided computer time.

## Author contributions

C.A.C. carried out the computations, experiments, and initial analysis; Y.P.L. formulated the research project, finalize the analysis, and wrote the manuscript with contributions from C.A.C.

## Competing interests

The authors declare no competing interests.
