## [Peer Review File · Communications Chemistry]

REVIEWERS' COMMENTS:

Reviewer #1 (Remarks to the Author):

The manuscript described the method of producing MVKO and characterized the syn-trans- and syn-cis-conformers with transient infrared spectra recorded using a step-scan Fourier-transform spectrometer. Coupling the theoretical calculations to identify the absorption peaks of the experimental infrared spectrum. This work gives a fairly complete set of detection methods. The manuscript can be accepted after minor revisions.

Some minor question were listed below:

1. The method of optimization and the theoretical prediction of infrared spectroscopy. Why B3LYP/aug-cc-pVTZ-pp was chosen as the method to optimize molecular structure and the theoretical FT-IR spectroscopy, not the method used in the author's article published in 2013?
2. A correction factor is usually used in infrared absorption spectrum prediction by theoretical using different theories. Is the theoretical infrared spectral peak position corrected in this paper? And the corrected factors are different among high-frequency zone and low-frequency zone. How do you choose the factors?
3. How to explain CH₂OO, syn-CH₃CHOO, anti-CH₃CHOO, (CH₃)₂COO, and C₂H₃C(CH₃)OO's O-O vibration absorption peaks located at 908, 871, 884, 887, and 948 cm⁻¹, respectively? And why only one CH₂ difference make O-O absorption peaks shifted so different? And there is no regular change of the position between those CIs. Is it possible that the actual peak position of O-O vibration absorption in CH₂OO locates at 848 cm⁻¹?

Reviewer #2 (Remarks to the Author):

Referee report for paper by C-A Chung and Y-P Lee on IR spectroscopy of resonantly stabilized Criegee intermediates.

The group of YP Lee has established the IR detection of small and medium sized Criegee intermediates, first based on the mechanism developed in the Taatjes group at SANDIA published in 2012. This breakthrough opened the view on an important part of atmospheric chemistry: It paves the way to a precise mechanistic level understanding of the gas phase chemistry of Criegee intermediates. The IR results of Lee and coworkers are based on the step scan technique and are highly complementary to other studies, because direct IR absorption provides the true absorption cross sections but also the rotational signature and thus a very precise picture of the intermolecular interactions.

In this paper for the first time a resonantly stabilized CI has been characterized, applying state of the art step scan IR spectroscopy and state of the art quantum chemistry. A key result is the difference in intensity as a measure for the increased stability of the resonantly stabilized O-O bond and its stretching motion! Normally IR intensities are not a good measure for the stability of chemical bonds because rigidity and sloppiness can both induce a steep gradient along the normal-coordinate (in first approximation) during the vibrational motion. This result alone makes this contribution a very important paper.

In addition, Chung and Lee show how the applied precursor chemistry (started by 1,3-diiodo-but-2-ene photolysis at 248 nm) for CI formation works by the detection of photolysis product C₂H₃C(CH₃)I.

I have just 2 minor comments/questions:

1)

Is the high level anharmonic quantum chemistry approach necessary? Would not be scaling say by

0.97 of harmonic frequencies sufficient? (see e.g. the very good agreement Fig. 4 c) in Angew. Chem. Int. Ed. 53 (2014) 715-719, where B3LYP/6-311+G(2d,2p) with 0.97 scaling was applied to calculate the spectra of Criegee intermediates and different secondary ozonides for beta-Pinene ozonolysis). Probably are the computational costs no longer such a big problem, but it also very important to check the real progress in accuracy, and to find better fittings for lower level methods which then can be applied to much larger systems!!

2)

Can you give an upper limit for the contribution of the other possible photolysis product (Z)-(CH₂I)CHC(CH₃)? This could be important for applying similar diiodo-compounds as precursors for larger unsaturated Criegee intermediates.

So I can recommend publication of the paper, but some discussion of the points I have risen I would highly appreciate.

Thomas Zeuch (Universität Göttingen)

Reviewer #3 (Remarks to the Author):

The paper 'Infrared characterization of resonance stabilized Criegee intermediates methyl vinyl ketone oxides C₂H₃C(CH₃)OO and its formation' by Chen-An Chung and Yuan-Pern Lee is a thorough and well conducted investigation. Generation of the MACRO Criegee intermediates is not a simple task and characterisation of the various stereoisomers requires careful analysis. The authors have interpreted the IR spectra with the assistance of theoretical studies and spectroscopic information derived will be of great help to further studies of these intermediates (kinetics etc.).

Therefore, I have no fundamental issues with the paper.

I would like to explore the formation of the Iodoperoxy radical with the authors. The \sim pressure observed was around 350 Torr? but presumably this is not the high pressure limit? wouldn't the iodoperoxy be expected to form at lower pressure? It would be useful. given the calculations already performed, to provide an indication of the expected low pressure regime, this would be helpful for experimental studies that follow.

Reviewer #4 (Remarks to the Author):

This manuscript describes infrared spectroscopy of methyl vinyl ketone oxide, a conjugated carbonyl oxide Criegee intermediate. The scientific importance of this work lies in at least three key areas: the confirmation that particular conformers of methyl vinyl ketone oxide are formed in the photolytic synthesis method; the characterization of the photolytic intermediates and the peroxy radical side products; the spectroscopic effects of resonance stabilization in the methyl vinyl ketone oxide. The measurements are solid, and the paper will be a valuable addition to the literature on these tropospherically important species. However, a little more detail and clarification could make the paper more effective.

First, the infrared spectra are compared to simulations built on DFT calculations, and that comparison forms the basis for identifying the conformers that are produced. It would be helpful to know the expected uncertainties in transition frequency and intensities for the DFT calculations and simulations and what quantitative methods were used to identify the spectral carrier. The two trans conformers have superficially very similar spectra, so is the spectrum assigned partly on knowledge that the anti-trans is unstable? When the authors state that the "observed spectrum

matches best with the spectrum simulated for syn-trans-MVKO" it would be good to have a description of how this best match is defined and measured, and to what extent this match excludes the possibility of anti- conformers contributing.

Second, the evidence of the effects of resonance stabilization is one of the most interesting aspects of this work, in my opinion. Can the authors give a qualitative argument as to why the resonance stabilization strengthens the O-O bond? I had reasonable success convincing myself based on drawing different resonance structures, so I think it can probably be done – that would make the connection stronger for many readers.

Finally, some small suggestions:

- the temporal behavior of different components – figure S13 – is not visible in the pdf version of the supplemental information but o.k. in the Word version.
- It would be useful to state the significance of the * symbol in the caption of Fig. 2

Response/revisions to the reviewers' comments

We appreciate very much the valuable comments and suggestions from the reviewers. Below are the detailed responses to the reviewer's comments about the manuscript. The reviewer comments are given in black, our responses are listed in blue color after each comment, and the revised text are highlighted in yellow.

Reviewer #1 (Remarks to the Author):

The manuscript described the method of producing MVKO and characterized the syn-trans- and syn-cis-conformers with transient infrared spectra recorded using a step-scan Fourier-transform spectrometer. Coupling the theoretical calculations to identify the absorption peaks of the experimental infrared spectrum. This work gives a fairly complete set of detection methods. The manuscript can be accepted after minor revisions. Some minor question were listed below:

1. The method of optimization and the theoretical prediction of infrared spectroscopy. Why B3LYP/aug-cc-pVTZ-pp was chosen as the method to optimize molecular structure and the theoretical FT-IR spectroscopy, not the method used in the author's article published in 2013?

Response/Revision: We employed the B3LYP/aug-cc-pVTZ method partly because it is more economic (the number of atoms increased from 5 in CH₂OO to 12 in MVKO) and partly because the results from B3LYP/aug-cc-pVTZ, especially the anharmonic vibrational calculations, are quite similar to the more sophisticated method in our paper in 2013.

2. A correction factor is usually used in infrared absorption spectrum prediction by theoretical using different theories. Is the theoretical infrared spectral peak position corrected in this paper? And the corrected factors are different among high-frequency zone and low-frequency zone. How do you choose the factors?

Response/Revision: For anharmonic vibrational wavenumbers, there is no need for correction. For harmonic vibrational calculations, we employed the equation $y = (0.9708 \pm 0.0159)x + (9.3 \pm 20.7)$, in which x is the harmonic vibrational wavenumbers. As stated in Supplementary Note 2, this equation was derived on fitting a linear plot of the experimental wavenumbers versus harmonic vibrational wavenumbers of (**1**) predicted with the B3LYP/aug-cc-pVTZ-pp method. In this work, the spectral region only covers 850–1450 cm⁻¹, so only one scaling equation was used.

3. How to explain CH₂OO, syn-CH₃CHOO, anti-CH₃CHOO, (CH₃)₂COO, and C₂H₃C(CH₃) OO's O-O vibration absorption peaks located at 908, 871, 884, 887, and 948 cm⁻¹,

respectively? And why only one CH₂ difference make O-O absorption peaks shifted so different? And there is no regular change of the position between those CIs. Is it possible that the actual peak position of O-O vibration absorption in CH₂OO locates at 848 cm⁻¹? **Response/Revision:** We have added a few sentences in page 7 of the main text to explain this: “Going from CH₂OO to *anti*-CH₃CHOO and (CH₃)₂COO, upon the replacement of the H atom(s) with CH₃, the OO-stretching wavenumber decreased slightly because of the electron-donating character of the CH₃ group enhances the C–O bond and weakens the O–O bond slightly. The OO-stretching wavenumber of *syn*-CH₃CHOO decreases further because of the interaction of the terminal O atom with the two H atoms in CH₃ weakens the O–O bond. The much larger OO-stretching wavenumber of MVKO is due to the resonance stabilization. The resonance stabilization can be comprehended from two major resonance structures shown in Supplementary Figure 9(a); the frontier orbitals of delocalized electrons with 0 to 2 nodes are shown in Supplementary Figure 9(b).”

No, the 848 cm⁻¹ is not the OO-stretching band because we have recorded high-resolution (0.002 cm⁻¹) spectrum of the band near 908 cm⁻¹ to show that its rotational structure fits well with predictions of a OO-stretching mode; please see DOI: 10.1039/c8cp04780d, *Phys. Chem. Chem. Phys.* **20**, 25806 (2018).

Reviewer #2 (Remarks to the Author):

Referee report for paper by C-A Chung and Y-P Lee on IR spectroscopy of resonantly stabilized Criegee intermediates.

The group of YP Lee has established the IR detection of small and medium sized Criegee intermediates, first based on the mechanism developed in the Taatjes group at SANDIA published in 2012. This breakthrough opened the view on an important part of atmospheric chemistry: It paves the way to a precise mechanistic level understanding of the gas phase chemistry of Criegee intermediates. The IR results of Lee and coworkers are based on the step scan technique and are highly complementary to other studies, because direct IR absorption provides the true absorption cross sections but also the rotational signature and thus a very precise picture of the intermolecular interactions.

In this paper for the first time a resonantly stabilized CI has been characterized, applying state of the art step scan IR spectroscopy and state of the art quantum chemistry. A key result is the difference in intensity as a measure for the increased stability of the resonantly stabilized O-O bond and its stretching motion! Normally IR intensities are not a good measure for the stability of chemical bonds because rigidity and sloppiness can both induce a steep gradient along the normal-coordinate (in first approximation) during the vibrational motion. This result alone makes this contribution a very important paper.

In addition, Chung and Lee show how the applied precursor chemistry (started by 1,3-diiodo-but-2-ene photolysis at 248 nm) for CI formation works by the detection of photolysis product C₂H₃C(CH₃)I.

I have just 2 minor comments/questions:

1)

Is the high level anharmonic quantum chemistry approach necessary? Would not be scaling say by 0.97 of harmonic frequencies sufficient? (see e.g. the very good agreement Fig. 4 c) in *Angew. Chem. Int. Ed.* 53 (2014) 715-719, where B3LYP/6-311+G(2d,2p) with 0.97 scaling was applied to calculate the spectra of Criegee intermediates and different secondary ozonides for beta-Pinene ozonolysis). Probably are the computational costs no longer such a big problem, but it also very important to check the real progress in accuracy, and to find better fittings for lower level methods which then can be applied to much larger systems!!

Response/Revision: We employed also the scaled harmonic vibrational wavenumbers and compared them in Supplementary Table 12. From this Table, the average absolute deviation between experiments and scaled harmonic was $16.5 \pm 10.8 \text{ cm}^{-1}$, with the maximal deviation of 32 cm^{-1} , whereas that for anharmonic vibrational calculations is $12.7 \pm 7.9 \text{ cm}^{-1}$, with a maximal deviation of 23 cm^{-1} . The anharmonic vibrational computations did provide slightly better agreements, but the scaled harmonic vibrational calculations are also sufficient. Because we intended to distinguish spectral patterns of various conformers, the small improvements by doing anharmonic vibrational calculations might help. The computational costs for DFT anharmonic vibrational calculations (without I atoms) are still acceptable, as the reviewer pointed out.

To address this more clearly, we have revised the related sentence on page 11–12 to “The anharmonic vibrations were calculated **only** for isomers of MVKO with a second-order perturbation approach using an effective finite-difference evaluation of the third and semi-diagonal fourth derivatives; **MVKO contains no I atom so that anharmonic vibrational calculations are feasible for molecules of this size.**”

We have also added one sentence on page 6 to show the differences: “**The average absolute deviation between experiments and anharmonic vibrational calculations of *syn-trans*-MVKO (3), $12.7 \pm 7.9 \text{ cm}^{-1}$, is much smaller than those for *syn-trans*-, *syn-cis*-, *anti-trans*-, and *anti-cis*-MVKO, 23.6 ± 13.7 , 23.4 ± 18.3 , and $25.4 \pm 16.5 \text{ cm}^{-1}$, , respectively; the listed errors represent one standard deviation in fitting. The corresponding average deviations for scaled harmonic vibrational calculations are 16.6 ± 10.8 , 24.9 ± 14.2 , 25.9 ± 18.5 , and $33.0 \pm 28.7 \text{ cm}^{-1}$, respectively.”**

2)

Can you give an upper limit for the contribution of the other possible photolysis product (Z)-(CH₂I)CHC(CH₃)? This could be important for applying similar diiodo-compounds as precursors for larger unsaturated Criegee intermediates.

Response/Revision: We could not estimate the upper limit for the branching of the formation of (Z)-(CH₂I)CHC(CH₃) because, in region 1450–850 cm⁻¹, the only intense line predicted for this species is near 1146 cm⁻¹, which might overlap with the intense band of the precursor near 1152 cm⁻¹; the intensities of other bands are less than 13 % of this band. However, from our work in solid *p*-H₂, J. Phys. Chem. 124, 5887 (2020), which covers a broader spectral range, we observed no trace of (Z)-(CH₂I)CHC(CH₃) after irradiation of the precursor at 280 nm. To this point clearer, we moved part of the Supplementary Note 3, including (now) Figure 2, to the main text. The paragraph on page 4–5 now reads:

“The precursor 1,3-diiodo-but-2-ene (CH₂I)HC=C(CH₃)I is predicted to exist in (Z)- and (E)-conformations, with the former (**1**) 7.3 kJ mol⁻¹ more stable than the latter. We employed pure (Z)-conformer in this experiment; its IR spectral characterization is described in Supplementary Note 2 and shown in Supplementary Figure 5. When the diiodoalkene precursor (**1**) in N₂ was irradiated with light at 248 nm, we observed six features near 1406, 1261, 1109, 1019, 925, and 873 cm⁻¹, as discussed in Supplementary Note 3 and shown in Supplementary Figure 6. We termed these six features that are associated with the primary photolysis product as group A and marked them A₁–A₆ in Figure 2(a). The stick spectra of two possible photolysis products, (Z)-C₂H₃C(CH₃)I (**2**) and (Z)-(CH₂I)CHC(CH₃), according to the scaled harmonic vibrational wavenumbers predicted with the B3LYP method are shown in Figures 2(b) and 2(c), respectively. The observed new features agree satisfactorily with lines predicted near 1418, 1261, 1108, 1018, 930, and 887 cm⁻¹ for (**2**), as compared in Figure 2 and Supplementary Table 11; they agree poorly with the spectrum predicted for (Z)-(CH₂I)CHC(CH₃), shown in Figure 2(c), and (E)-C₂H₃C(CH₃)I, shown in Supplementary Figure 7. Observation of (Z)-C₂H₃C(CH₃)I confirms that the terminal C–I bond was broken upon irradiation at 248 nm and that an interconversion between the (Z)- and (E)-conformers did not occur. We could not, however, definitely exclude the formation of (Z)-(CH₂I)CHC(CH₃) in a small proportion because, in region 1450–850 cm⁻¹, the only intense line predicted for this species is near 1146 cm⁻¹ (Figure 2c), which might overlap with the intense band of the precursor near 1152 cm⁻¹; hence the upper limit for the percentage of production of (Z)-(CH₂I)CHC(CH₃) could not be estimated. However, according to our previous results on photolysis of (Z)-(CH₂I)HC=C(CH₃)I in solid *p*-H₂ at 290 nm, which cover the entire IR region to include several other intense features of the products, the formation of (Z)-(CH₂I)CHC(CH₃) on breaking the central C–I bond was unobserved.³²”

So I can recommend publication of the paper, but some discussion of the points I have risen I would highly appreciate.

Thomas Zeuch (Universität Göttingen)

Reviewer #3 (Remarks to the Author):

The paper 'Infrared characterization of resonance stabilized Criegee intermediates methyl vinyl ketone oxides $C_2H_3C(CH_3)OO$ and its formation' by Chen-An Chung and Yuan-Pern Lee is a thorough and well conducted investigation. Generation of the MACRO Criegee intermediates is not a simple task and characterisation of the various stereoisomers requires careful analysis. The authors have interpreted the IR spectra with the assistance of theoretical studies and spectroscopic information derived will be of great help to further studies of these intermediates (kinetics etc.).

Therefore, I have no fundamental issues with the paper.

I would like to explore the formation of the Iodoperoxy radical with the authors. The ~ pressure observed was around 350 Torr? but presumably this is not the high pressure limit? wouldn't the iodoperoxy be expected to form at lower pressure? It would be useful, given the calculations already performed, to provide an indication of the expected low pressure regime, this would be helpful for experimental studies that follow.

Response/Revision: As stated in Supplementary Notes 6 and 8, the pressure for the more enhanced production of iodoperoxy radical is 246 and 347 Torr, but some iodoperoxy radical still exists at 35 and 82 Torr (Supplementary Table 15). From our experience with CH_2OO , the branching between CH_2OO and CH_2IOO is continuously changing with pressure; please see *J. Phys. Chem. Lett.* **6**, 4610–4615 (2015). The 347 Torr we employed is not the high-pressure limit, as we still observed some MVKO at the beginning. The reason that we did not perform experiments at higher pressure was because experiments became less feasible as the flow might become more turbulent so that the noise increased significantly; furthermore, the significantly increased sample consumption would become unaffordable. The determination of the branching ratio between MVKO and the peroxy adduct as a function of pressure requires significant efforts and will be carried out in the future. We have added one paragraph on P. 9 before Conclusion to read “Based on temporal profiles of varied species during the reaction, one would be able to obtain information such as rate coefficients of formation, the relative yields of MVKO (3) and $C_2H_3C(CH_3)IOO$ (4) as a function of pressure, and kinetics on reactions of MVKO (3) with atmospheric species. More careful experiments under varied conditions are needed.”

Reviewer #4 (Remarks to the Author):

This manuscript describes infrared spectroscopy of methyl vinyl ketone oxide, a conjugated carbonyl oxide Criegee intermediate. The scientific importance of this work lies in at least three key areas: the confirmation that particular conformers of methyl vinyl ketone oxide are formed in the photolytic synthesis method; the characterization of the photolytic intermediates and the peroxy radical side products; the spectroscopic effects of resonance stabilization in the methyl vinyl ketone oxide. The measurements are solid, and the paper will be a valuable addition to the literature on these tropospherically important species. However, a little more detail and clarification could make the paper more effective.

First, the infrared spectra are compared to simulations built on DFT calculations, and that comparison forms the basis for identifying the conformers that are produced. It would be helpful to know the expected uncertainties in transition frequency and intensities for the DFT calculations and simulations and what quantitative methods were used to identify the spectral carrier. The two trans conformers have superficially very similar spectra, so is the spectrum assigned partly on knowledge that the anti-trans is unstable? When the authors state that the “observed spectrum matches best with the spectrum simulated for syn-trans-MVKO” it would be good to have a description of how this best match is defined and measured, and to what extent this match excludes the possibility of anti-conformers contributing.

Response/Revision: The average of absolute deviations between experiments and calculations are 12.7 ± 7.9 , 23.6 ± 13.7 , 23.4 ± 18.3 , and 25.4 ± 16.5 cm^{-1} for *syn-trans*-, *syn-cis*-, *anti-trans*-, and *anti-cis*-MVKO, respectively; furthermore, the rotational contours for each band varies with the type of transition (see Figure 3), especially for c-type bands, and the agreement is the best for *syn-trans*-MVKO. We hence stated that the observed spectrum matches best with the spectrum simulated for *syn-trans*-MVKO. We have revised the paragraph on pages 5–6 to be: “The observed spectrum matches best with the spectrum simulated for *syn-trans*-MVKO (**3**) in terms of vibrational wavenumbers, relative intensities, and rotational contours. The observed vibrational wavenumbers near 1416, 1383, 1346, 1060, 987, 948, and 908/916 cm^{-1} agree satisfactorily with the anharmonic vibrational wavenumbers predicted near 1420, 1396, 1365, 1048, 1004, 947, and 939 cm^{-1} for (**3**), as compared in Supplementary Table 12. The average absolute deviation between experiments and anharmonic vibrational calculations of *syn-trans*-MVKO (**3**), 12.7 ± 7.9 cm^{-1} , is much smaller than those for *syn-trans*-, *syn-cis*-, *anti-trans*-, and *anti-cis*-MVKO, 23.6 ± 13.7 , 23.4 ± 18.3 , and 42.6 ± 46.8 cm^{-1} , respectively. The corresponding average deviations for scaled harmonic vibrational calculations are 16.6 ± 10.8 , 24.9 ± 14.2 , 25.9 ± 18.5 , and 33.0 ± 28.7 cm^{-1} , respectively.”

Second, the evidence of the effects of resonance stabilization is one of the most interesting aspects of this work, in my opinion. Can the authors give a qualitative argument as to why the resonance stabilization strengthens the O-O bond? I had reasonable success convincing myself based on drawing different resonance structures, so I think it can probably be done – that would make the connection stronger for many readers.

Response/Revision: We have added the resonance structures in Supplementary Figure 9(a) to show the stabilization of the O–O bond and in Supplementary Figure 9(b) to show the delocalized Frontier orbitals. The text on page 7 has been revised as “The resonance stabilization can be comprehended from two major resonance structures with conjugated double bonds, shown in Supplementary Figure 9(a); the frontier orbitals of delocalized electrons with 0 to 2 nodes are shown in Supplementary Figure 9(b).”

Finally, some small suggestions:

- the temporal behavior of different components – figure S13 – is not visible in the pdf version of the supplemental information but o.k. in the Word version.

Response/Revision: We have checked the pdf file to make sure that all figures turn out fine.

- It would be useful to state the significance of the * symbol in the caption of Fig. 2

Response/Revision: We added one sentence in the caption: “Possible bands of *syn-cis*-MVKO are marked with blue asterisks.”

Reviewer #1 (Remarks to the Author):

The authors had revised their manuscript carefully, now it is acceptable as it is.

Reviewer #2 (Remarks to the Author):

The authors have thoroughly addressed the comments of the reviewers. I recommend to publish the paper as is.

Thomas Zeuch